Acute kidney injury-attributable mortality in critically ill patients with sepsis

Wang Zhiyi 1 2
Weng Jie 1
Yang Jinwen 3
Zhou Xiaoming 1
Xu Zhe 4
Hou Ruonan 1
Zhou Zhiliang 1
Wang Liang 5
Chen Chan 3
Jin Shengwei jinshengwei69@163.com 6
1 Department of General Practice, The Second Affiliated Hospital and Yuying Children’s Hospital of Wenzhou Medical University , Wenzhou , China
2 Center for Health Assessment, Wenzhou Medical University , Wenzhou , Zhejiang , China
3 Department of Geriatric Medicine, The First Affiliated Hospital, Wenzhou Medical University , Wenzhou , China
4 Department of Emergency Intensive Care Unit, The Second Affiliated Hospital and Yuying Children’s Hospital of Wenzhou Medical University , Wenzhou , China
5 Department of Public Health, Robbins College of health and Human Sciences, Baylor University , Waco , TX , United States of America
6 Department of Anesthesia and Critical Care, the Second Affiliated Hospital of Wenzhou Medical University , Wenzhou , China
Capusa Cristina
Electronic publication date: 2022 Mar 25
Publication date: 2022
Volume: 10
Electronic Location ID: e13184
Received 2022 Jan 11; Accepted 2022 Mar 7
Copyright: ©2022 Wang et al.
Copyright year: 2022
Copyright holder: Wang et al.
License: This is an open access article distributed under the terms of the Creative Commons Attribution License, which permits unrestricted use, distribution, reproduction and adaptation in any medium and for any purpose provided that it is properly attributed. For attribution, the original author(s), title, publication source (PeerJ) and either DOI or URL of the article must be cited.
License URL: https://creativecommons.org/licenses/by/4.0/

Keywords: Acute kidney injury, Attributable mortality, Sepsis, Mortality

Funding: National Natural Science Foundation of China 81772054, 82100074 The Key R&D Program of Zhejiang Province 2019C03011 Wenzhou Major Science and Technology Innovation Project 2018ZY006 Clinical Research Fundation of the 2nd Affiliated Hospital of Wenzhou Medical University SAHoWMU-CR2018-11-134 This study was supported by National Natural Science Foundation of China, No. 81772054 and 82100074. The Key R&D Program of Zhejiang Province (2019C03011), Wenzhou Major Science and Technology Innovation Project (2018ZY006), Clinical Research Fundation of the 2nd Affiliated Hospital of Wenzhou Medical University (SAHoWMU-CR2018-11-134). The funders had no role in study design, data collection and analysis, decision to publish, or preparation of the manuscript.

==============================
Background

To assess whether acute kidney injury (AKI) is independently associated with hospital mortality in ICU patients with sepsis, and estimate the excess AKI-related mortality attributable to AKI.

Methods

We analyzed adult patients from two distinct retrospective critically ill cohorts: (1) Medical Information Mart for Intensive Care IV (MIMIC IV; n = 15,610) cohort and (2) Wenzhou (n = 1,341) cohort. AKI was defined by Kidney Disease: Improving Global Outcomes (KDIGO) criteria. We applied multivariate logistic and linear regression models to assess the hospital and ICU mortality, hospital length-of-stay (LOS), and ICU LOS. The excess attributable mortality for AKI in ICU patients with sepsis was further evaluated.

Results

AKI occurred in 5,225 subjects in the MIMIC IV cohort (33.5%) and 494 in the Wenzhou cohort (36.8%). Each stage of AKI was an independent risk factor for hospital mortality in multivariate logistic regression after adjusting for baseline illness severity. The excess attributable mortality for AKI was 58.6% (95% CI [46.8%–70.3%]) in MIMIC IV and 44.6% (95% CI [12.7%–76.4%]) in Wenzhou. Additionally, AKI was independently associated with increased ICU mortality, hospital LOS, and ICU LOS.

Conclusion

Acute kidney injury is an independent risk factor for hospital and ICU mortality, as well as hospital and ICU LOS in critically ill patients with sepsis. Thus, AKI is associated with excess attributable mortality.

Introduction

Acute kidney injury (AKI) is a prevalent clinical complication among patients in Intensive Care Units (ICUs), and an independent risk factor for those in critical conditions (Barrantes et al., 2008; Nisula et al., 2013). Currently, there are no effective drugs available for AKI management (Peerapornratana et al., 2019). Studies have explored the database for critically ill patients and found that each stage of AKI is associated with high mortality (Joannidis et al., 2009; Khadzhynov et al., 2019; Li, Zou & Xu, 2016). Among ICU patients with liver cirrhosis, an analysis of matched population-based cohort revealed excess mortality attributable to severe AKI and mild AKI at 51% and 25%, respectively (Du Cheyron et al., 2005). Sepsis is the leading cause of AKI in critically ill patients (Peerapornratana et al., 2019). It is approximated that one-third of sepsis patients develop AKI (Murugan et al., 2010). Sepsis-associated AKI is a frequent complication in critically ill patients and contributes to high mortality (Peerapornratana et al., 2019; Poston & Koyner, 2019). Kellum et al. (2016) reported that sixty-day hospital mortality was 6.2% for septic Shock patients without AKI, 16.8% for those with stage 1, and 27.7% for stages 2–3. Currently, all available literature only reports sepsis-related AKI mortality (Chang et al., 2015; Uhel et al., 2020). However, the attributable mortality for AKI in ICU patients with sepsis is unknown. Assessment of the AKI attributable mortality would guide in designing clinical trials for the prevention or treatment of AKI.

This study aims to assess whether the development of AKI is an independent risk factor for mortality in ICU patients with sepsis and to adequately evaluate the excess mortality attributable to AKI.

Materials & Methods

Participants

Critically ill adult patients were enrolled from two distinct retrospective ICU cohorts: (1) Medical Information Mart for Intensive Care IV (MIMIC IV) cohort and (2) Wenzhou cohort study (Zhou et al., 2021). The MIMIC IV cohort was enrolled from a relational database containing comprehensive information on over 250,000 patients hospitalized between 2008 and 2019 at Beth Israel Deaconess Medical Center in Boston, MA, USA. The Wenzhou cohort included critically ill adult patients from ICUs at the Second Affiliated Hospital of Wenzhou Medical University in Wenzhou, Zhejiang, China. The MIMIC IV public database was approved by the institutional review board (IRB). Wenzhou cohort was approved by the Second Affiliated Hospital of Wenzhou Medical University IRB. Informed consent was waived due to retrospective nature of the study.

Patients

Inclusion criteria for adult patients followed the Third International Consensus Definitions for Sepsis and Septic Shock (Sepsis-3) i.e., a known or suspected infection plus acute increase Sequential Organ Failure Assessment (SOFA) ≥ 2 points for organ dysfunction (Shankar-Hari et al., 2016; Singer et al., 2016) from the MIMIC IV and Wenzhou cohorts. We excluded patients with a history of chronic kidney disease (glomerulonephritis, diabetic nephropathy, hypertensive nephropathy, hereditary nephritis, and chronic kidney failure caused by a variety of other diseases), multiple hospitalizations (only the first hospitalization was considered), and ICU length of stay (LOS) less than 24 h. In both cohorts, the SOFA score (Vincent et al., 1996), Acute Physiology Score (APS) III (Knaus et al., 1991), Logistic Organ Dysfunction Score (LODS) (Le Gall et al., 1996), and Oxford Acute Severity of Illness Score (OASIS) (Johnson, Kramer & Clifford, 2013) were employed to evaluate the severity of illness. Calculations for the modified SOFA score, modified APS III and modified LODS were obtained through the exclusion of points associated with renal function. The shock and respiratory failure variables were both collected. We defined shock as the need for vasopressor within the first 48 h of hospital admission, while respiratory failure was defined as the need for invasive mechanical ventilation.

Outcomes

AKI is the primary outcome. Patients were defined as having AKI if they met the Kidney Disease: Improving Global Outcomes (KDIGO) serum creatinine diagnostic criteria for AKI (Supplemental Information) (Kellum et al., 2012). The presence of AKI and its severity were defined according to KDIGO criteria. The secondary outcomes included hospital and ICU mortality, hospital LOS and ICU LOS.

Statistical methods

Wilcoxon rank-sum test, Student’s t-test, and Chi-squared test were employed to compare the baseline characteristic variables. Before data analysis, the potential confounders and mediating variables between AKI and death were depicted in the directed acyclic graph (DAG) (Fig. 1) (Lederer et al., 2019). Potential confounders are variables related to both the exposure and outcome of interest, and therefore should be controlled for in analyses. Variables that may be on the indirect causal path between the exposure and outcome partially mediate the association between exposure and outcome. It would be incorrect to control for these factors, as that would partially close the causal path, attenuating the observed association between the exposure and outcome (Lederer et al., 2019).

Figure 1 Direct acyclic graph for the relationship between AKI and death.

Guided by the conceptual model illustrated in the DAG, multivariate logistic, and linear binomial regression models were developed for primary and secondary outcomes. Specifically, we assessed each of the variables identified on the DAG and listed in Table 1 for inclusion in the final multivariable model along with several pre-specified variables considered clinically significant pre-hoc. Pre-specified variables included illness severity score (modified APS III, SOFA score, modified LODS and OASIS), age, gender, race, and shock. Chronic co-morbidity variables for the final model were selected after assessing how each potential confounder affected the overall odds ratio (OR) when added to a multivariable model with the pre-specified variables. In generating final models, we retained only variables that altered the OR by greater than 10%, so as to avoid over-fitting.

Table 1 Variables screened for inclusion into models of attributable mortality.

Pre-specified variables for inclusion in Models	
Age	
Baseline severity of illness (modified APS III, modified SOFA, modified LODS, OASIS)	
Demographics	
Gender	
Race*	
Chronic Co-morbidities	
Heart disease (myocardial infarct, congestive heart failure)	
Liver disease	
Chronic pulmonary disease	
Rheumatic disease	
Diabetes	
Cancer (metastatic cancer, non-metastatic cancer)	
AIDS	
Cerebrovascular disease	
Acute Co-morbidities	
Shock (use of vasopressors)	
Respiratory failure (receive mechanical ventilation)	
Notes.

* Variables only considered within MIMIC IV cohort, not present within Wenzhou cohort.

We performed a sensitivity analysis of hospital and ICU LOS for all patients (both survivors and non-survivors). Sepsis-associated AKI was re-defined according to the urine output diagnostic criteria of KDIGO for AKI (sensitivity analysis with urine output criteria, considered only urine output criteria, not the worse between serum creatinine and urine output criteria) (Supplemental Information). Furthermore, sensitivity analyses of hospital and ICU mortality were conducted based on the urine output diagnostic criteria.

The attributable fraction (AF) of mortality from AKI (A FAKI) and the population AF of mortality from AKI (population A FAKI) were calculated as reported previously (detail for this calculation is provided in Supplemental Information) (Auriemma et al., 2020; Van Vught et al., 2016). The A FAKI denoted the proportion of deaths attributable to AKI in septic patients with AKI. Population A FAKI denoted the proportion of all deaths in the sepsis population attributable to AKI. Estimated value was generated by indirect standardization, performed within strata (additional details for this calculation are provided in Supplemental Information). All statistical analyses were conducted in R (version 3.6.1) used in our previous study (Weng et al., 2021; Xu et al., 2021); p-value < 0.05 denoted statistical significance.

Results

Baseline characteristics and outcomes

Figure 2 illustrates patient selection flow chart, whereas Table 2 outlines the baseline patient characteristics. The Wenzhou cohort tended to be older with higher vasopressor use probability compared to those of the MIMIC IV cohort. The baseline modified SOFA score, modified APS III, modified LODS and OASIS were similar between the two cohorts. The proportion of patients requiring continuous renal replacement therapy (CRRT) were similar, however, more patients acquired AKI in the Wenzhou cohort compared to the MIMIC IV cohort. We reported more cases of stage 1 AKI in the Wenzhou population compared to the MIMIC IV population. While hospital and ICU LOS were longer in the Wenzhou cohort, hospital and ICU mortalities were higher in the MIMIC IV cohort.

Figure 2 Study flowcharts for the MIMIC IV and Wenzhou cohorts.

Table 2 Baseline characteristics of MIMIC IV and Wenzhou cohorts, together and stratified by AKI.

Clinical variable*	All patients (n= 16,951)	MIMIC IV (n= 15,610)	Wenzhou (n= 1,341)	
	MIMIC IV (n= 15,610)	Wenzhou (n = 1,341)	P value	No AKI (n = 10,385)	AKI (n = 5,225)	P value	No AKI (n = 847)	AKI (n = 494)	P value	
Age, years	64 ± 17	69 ± 10	<0.001	63 ± 17	64 ± 16	0.27	70 (63, 76)	70 (62, 75)	0.617	
Male gender, %	8,889 (57)	778 (58)	0.464	5,863 (56)	3,026 (58)	0.086	489 (58)	289 (59)	0.828	
White race, %	10,537 (68)	–	–	7,143 (69)	3,394 (65)	<0.001	–	–	–	
APS III	46 (33, 66)	45 (33, 65)	0.447	40 (31, 54)	63 (44, 87)	<0.001	40 (31, 53)	60 (42, 81)	<0.001	
Modified APS IIIa	36 (27, 53)	35 (27, 51)	0.151	33 (25, 44)	49 (33, 71)	<0.001	32 (25, 43)	44.5 (32, 66)	<0.001	
SOFA score	5 (4, 8)	5 (4, 8)	0.605	5 (3, 7)	8 (5, 11)	<0.001	4 (3, 6)	7 (5, 11)	<0.001	
Modified SOFA scorea	5 (3, 7)	5 (3, 7)	0.57	4 (3, 6)	7 (4, 10)	<0.001	4 (3, 6)	6 (4, 9)	<0.001	
LODS	5 (3, 7)	5 (3, 7)	0.901	4 (2, 6)	7 (5, 10)	<0.001	4 (2, 6)	7 (4, 9)	<0.001	
Modified LODSa	3 (1, 5)	3 (1, 5)	0.732	2 (1, 4)	5 (2, 7)	<0.001	2 (1, 4)	4 (2, 6.75)	<0.001	
OASIS	34 (28, 40)	34 (28, 40)	0.827	32 (26, 37)	38 (32, 45)	<0.001	32 (27, 38)	38 (31, 45)	<0.001	
Vasopressor use in first 48 h, %	7,529 (48)	708(53)	0.001	4,262 (41)	3,267 (63)	<0.001	387 (46)	321 (65)	<0.001	
Mechanical ventilation, %	11,334 (73)	986 (74)	0.488	6,840 (66)	4,494 (86)	<0.001	561 (66)	425 (86)	<0.001	
CRRT, %	561 (4)	45 (3)	0.708	26 (0)	535 (10)	<0.001	1 (0)	44 (9)	<0.001	
AKI, %	5,225 (33)	494 (37)	0.013	–	–	–	–	–	–	
Stage 1 AKI, %	3,187 (20)	323 (24)		–	–	–	–	–	–	
Stage 2 AKI, %	1,117 (7)	104 (8)		–	–	–	–	–	–	
Stage 3 AKI, %	921 (6)	67 (5)		–	–	–	–	–	–	
Hospital LOS	8 (5, 13)	9 (6, 14)	<0.001	7 (4, 11)	11 (6, 20)	<0.001	8 (6, 12)	12 (7, 19)	<0.001	
Hospital LOSb	8 (5, 13)	9 (6, 14)	<0.001	7 (4, 11)	12 (7, 22)	<0.001	8 (6, 12)	12 (7, 20)	<0.001	
ICU LOS	2 (1, 5)	4 (2, 6)	<0.001	2 (1, 3)	5 (2, 9)	<0.001	3 (2, 5)	5 (3, 10)	<0.001	
ICU LOSb	2 (1, 5)	4 (2, 6)	<0.001	2 (1, 3)	5 (2, 10)	<0.001	3 (2, 5)	5 (3, 10)	<0.001	
Hospital mortality, %	2,118 (14)	155 (12)	0.042	727 (7)	1,391 (27)	<0.001	51 (6)	104 (21)	<0.001	
ICU mortality, %	1,478 (9)	111 (8)	0.165	413 (4)	1,065 (20)	<0.001	26 (3)	85 (17)	<0.001	
Notes.

APS Acute Physiology Score

SOFA Sequential Organ Failure Assessment

LODS Logistic Organ Dysfunction Score

OASIS Oxford Acute Severity of Illness Score

CRRT Continuous Renal Replacement Therapy

AKI Acute Kidney Injury

LOS length of stay

* Data shown as mean ± standard deviation, median (interquartile range) or number (percent) as appropriate.

a Modified scores exclude points related to renal function.

b Restricted to survivor.

Table 2 shows participant characteristics stratified by AKI status. in both cohorts, patients with AKI demonstrated a greater need for mechanical ventilation, vasopressor use, CRRT, and higher illness severity scores than patients without AKI; they also were characterized by higher mortality and longer LOS.

Comparison of clinical outcomes adjusted for severity of illness

MIMIC IV

We reported overall hospital mortality of 13.5%; briefly, 2,118 of 15,610 patients died before discharge (Table 3). Nearly 66% of non-survivors developed AKI, whereas 28% of survivors developed AKI (p < 0.001). More patient characteristics were shown in Table 3.

Table 3 Patient characteristics stratified by in-hospital mortality, MIMIC IV and Wenzhou cohorts.

Clinical variable*	Survived (n = 13,492)	Died (n = 2,118)	p value	
MIMIC IV patient characteristics				
Age, years	64 (53, 76.25)	68 (57, 81)	<0.001	
Male gender, %	7,772 (58)	1,117 (53)	<0.001	
White race, %	9,295 (69)	1,242 (59)	<0.001	
APS III	43 (32, 59)	81 (60, 103)	<0.001	
Modified APS IIIa	34 (26, 48)	64 (46, 83)	<0.001	
SOFA score	5 (3, 7)	9 (6, 13)	<0.001	
Modified SOFA scorea	4 (3, 7)	8 (5, 11)	<0.001	
LODS	4 (3, 6)	9 (6, 12)	<0.001	
Modified LODSa	2 (1, 4)	6 (4, 8)	<0.001	
OASIS	32 (27, 38)	43 (36, 49)	<0.001	
Vasopressor use in first 48 h, %	6,179 (46)	1,350 (64)	<0.001	
Mechanical ventilation, %	9,483 (70)	1,851 (87)	<0.001	
CRRT, %	235 (2)	326 (15)	<0.001	
AKI, %	3,834 (28)	1,391 (66)	<0.001	
Hospital LOS	8 (5, 13)	6 (3, 13)	<0.001	
ICU LOS	2 (1, 5)	4 (2, 8)	<0.001	
Clinical variable*	Survived (n = 1186)	Died (n = 155)	p value	
Wenzhou patient characteristics				
Age, years	70 (62, 76)	72 (66, 76)	0.029	
Male gender, %	692 (58)	86 (55)	0.553	
APS III	43 (32, 59)	77 (55, 97.5)	<0.001	
Modified APS IIIa	34 (26, 48)	59 (42, 80.5)	<0.001	
SOFA score	5 (3, 7)	9 (6, 13)	<0.001	
Modified SOFA scorea	5 (3, 7)	8 (5, 11)	<0.001	
LODS	4 (3, 6)	9 (6, 11)	<0.001	
Modified LODSa	2 (1, 4)	6 (4, 8)	<0.001	
OASIS	33 (27, 39)	42 (34.5, 47.5)	<0.001	
Vasopressor use in first 48 h, %	611 (52)	97 (63)	0.012	
Mechanical ventilation, %	856 (72)	130 (84)	0.003	
CRRT, %	27 (2)	18 (12)	<0.001	
AKI, %	390 (33)	104 (67)	<0.001	
Hospital LOS	9 (6, 14)	8 (4.5, 15.5)	0.074	
ICU LOS	4 (2, 6)	5 (3, 8.5)	<0.001	
Notes.

* Data shown as mean ± standard deviation, median (interquartile range) or number (percent) as appropriate.

a Modified APACHE scores exclude points related to renal function.

The unadjusted hospital mortality of sepsis with AKI was 27%, while that for sepsis without AKI was 7% (Table 4; OR = 4.82; 95% CI [4.37–5.31]; p < 0.001). In constructing the adjusted model, no other variables except the prespecified variables (illness severity score, age, gender, race, and shock) met the set criteria. In the multivariable regression model, the OR values for hospital mortality of patients with AKI were attenuated but remained statistically significant after adjustment for illness severity score (modified APS III, SOFA score, modified LODS, and OASIS), age, gender, race, and shock. Sensitivity analyses based on the urine output diagnostic criteria of KDIGO for AKI yielded similar results (Table S1). Septic patients who developed AKI, experienced longer hospital and ICU LOS than patients without AKI, whether among survivors or across all patients (Tables S2 and S3).

Table 4 Association of AKI with mortality in unadjusted and adjusted models, MIMIC IV and Wenzhou cohorts.

MIMIC IV logistic regression models (n= 15,610)	OR (95% CI)	p value	
Unadjusted model of AKI for in-hospital mortality	4.82 (4.37, 5.31)	<0.001	
Adjusted for modified APS IIIa	2.57 (2.30, 2.87)	<0.001	
Adjusted for modified SOFA scorea	2.89 (2.59, 3.21)	<0.001	
Adjusted for modified LODSa	2.72 (2.44, 3.03)	<0.001	
Adjusted for OASIS	2.88 (2.58, 3.20)	<0.001	
Unadjusted model of AKI for ICU mortality	6.18 (5.49, 6.97)	<0.001	
Adjusted for modified APS IIIb	2.86 (2.51, 3.27)	<0.001	
Adjusted for modified SOFA scoreb	3.30 (2.90, 3.76)	<0.001	
Adjusted for modified LODSb	2.98 (2.61, 3.30)	<0.001	
Adjusted for OASIS	3.15 (2.77, 3.59)	<0.001	
Wenzhou logistic regression models (n= 1,341)	OR (95% CI)	p value	
Unadjusted model of AKI for in-hospital mortality	4.16 (2.93, 5.98)	<0.001	
Adjusted for modified APS IIIa	2.44 (1.65, 3.64)	<0.001	
Adjusted for modified SOFA scorea	2.64 (1.79, 3.91)	<0.001	
Adjusted for modified LODSa	2.48 (1.68, 3.68)	<0.001	
Adjusted for OASIS	2.75 (1.88, 4.07)	<0.001	
Unadjusted model of AKI for ICU mortality	6.56 (4.22, 10.53)	<0.001	
Adjusted for modified APS IIIb	3.45 (2.12, 5.75)	<0.001	
Adjusted for modified SOFA scoreb	3.85 (2.39, 6.37)	<0.001	
Adjusted for modified LODSb	3.38 (2.09, 5.62)	<0.001	
Adjusted for OASIS	3.96 (2.46, 6.53)	<0.001	
Notes.

Modified scores exclude points related to renal function.

a In addition to severity of illness variable listed in the table, adjusted models include age, gender, race, and shock.

b In addition to severity of illness variable listed in the table, adjusted models include age, gender and shock.

Furthermore, we conducted stratified analyses based on the severity of AKI. And found that stages 1, 2, and 3 AKI were all independently associated with hospital and ICU mortality in adjusted and unadjusted models (Fig. 3, Table S4). In four adjusted models, stage 3 AKI exhibited the most significant association with increased risk of hospital and ICU mortality. In the MIMIC IV cohort, the AFAKI was 58.6% (CI [46.8%–70.3%]), whereas the population AFAKI was 30.2% (95% CI [22.7%–37.8%]).

Figure 3 Odds ratios with 95% confidence intervals for in-hospital mortality stratified by severity of AKI.

In addition to severity of illness variables listed in the Figure, adjusted models for MIMIC IV include age, gender, race, and shock. Adjusted models for Wenzhou include age, gender and shock.

Wenzhou

Among 1,341 patients, 155 patients died before discharge, with overall hospital mortality of 10.3% (Table 3). Nearly 67% of non-survivors developed AKI, whereas 33% of survivors developed AKI (p < 0.001). More patient characteristics were shown in Table 3.

The unadjusted hospital mortality of sepsis with AKI was 21% while that for sepsis without AKI was 6% (Table 4; OR = 4.16; 95% CI [2.93–5.98]; p < 0.001). Similar to findings in the MIMIC IV cohort, development of AKI in the Wenzhou population was significantly associated with increased risk of hospital and ICU mortality in multivariate logistic regression after we adjusted for illness severity score (modified APS III, SOFA score, modified LODS and OASIS), age and shock. Similarly, in the sensitivity analyses, AKI was associated with ICU mortality, when we applied the urine output diagnostic criteria of KDIGO for AKI (Table S1).

As in the MIMIC IV cohort, patients with AKI had prolonged hospital and ICU (Tables S2 and S3). Moreover, the correlation of AKI with mortality was stratified according to the severity of AKI. In the Wenzhou cohort, stages 1, 2, and 3 AKI were independently associated with hospital and ICU mortality (Fig. 3, Table S4). In the Wenzhou cohort, the A FAKI was 44.6% (95% CI [12.7%–76.4%]), whereas the population A FAKI was 26.0% (95% CI [0%–56.8%]).

Discussion

We have revealed the association of AKI with mortality in two critically ill cohorts. Stages 1, 2, and 3 AKI were associated with a longer hospital and ICU LOS, as well as greater hospital and ICU mortality. It is not surprising that patients with stage 3 AKI are characterized by a worse prognosis than those with stages 1- and 2 AKI. Our results provide implicate AKI as an independent risk factor for mortality in patients with sepsis.

Lopes et al. (2010) demonstrated that AKI had a negative impact on long-term mortality of patients with sepsis. Uhel et al. (2020) reported persistent AKI is independently associated with sepsis mortality compared with transient AKI. Our study yielded similar results to previous studies. However, to our knowledge, this is the first study to explore the excess mortality attributable to AKI in septic patients with severe illness. We applied the KDIGO serum creatinine and urine output diagnostic criteria for AKI. whereas, the potential confounders and mediating variables between AKI and death were depicted in DAG. With this approach, we elucidated the relationship between variables and avoided including mediator variables.

Previous reports show that increased severity of AKI is correlated with a stepwise increase in mortality among critically ill patients (Panitchote et al., 2019; Uchino et al., 2006; Uchino et al., 2005), which concurred with our findings. Elsewhere. Vaara et al. (2014) reported that stage 1 AKI was not a substantial risk factor for 90-day mortality in critically ill patients. Through matched risk-adjusted mortality, Cheyron et al. found that only severe AKI was significantly associated with excess attributable mortality in ICU patients with liver cirrhosis (Du Cheyron et al., 2005). Herein, we have reported different results for hospital and ICU mortality compared to the results of Vaara et al. and Cheyron et al., which may be attributed to differences in severity of the disease and that we focused on critically ill patients with sepsis. Additionally, the development of AKI had been associated with long-term risk of mortality and other adverse outcomes, including chronic kidney disease (CKD) and end-stage renal disease (ESRD) (Coca et al., 2009; Fortrie, De Geus & Betjes, 2019). AKI occurrence was mostly in association with sepsis in critically ill patients. Currently, no effective cure or effective treatment is available yet and clinical interventions are limited (Al-Jaghbeer et al., 2018; Skube et al., 2018). Therefore, the prevention of sepsis-induced AKI is critical in reducing the case fatality rate.

Of note, we estimated the AFAKI and population AFAKI in two cohorts and yielded similar results. Few studies have assessed the attributable mortality of AKI. In one study of critically ill patients with liver cirrhosis, the AFAKI from mild AKI was 25% and the AFAKI from severe AKI was 51% (Du Cheyron et al., 2005), whereas the 90-day AFAKI in ICU patients was 8.6%, and population AFAKI was nearly 20% (Vaara et al., 2014) in another study. It is imperative to apply our results to estimate the attributable mortality of other critically ill patients. In ICU patients, the AF of mortality from sepsis was 15% (Shankar-Hari et al., 2018). The AF of ARDS in patients with sepsis was 27% and 37% in EARLI and VALID cohorts, respectively (Auriemma et al., 2020).

There are several highlights in the present study. First, we included two independent large cohorts of critically ill adult patients hospitalized with sepsis from two countries. The similarity of the association between AKI and mortality in two cohorts strengthens the validity and generalizability of our findings. Second, we reported consistent results we adjusted for four different severity of illness scores in two cohorts. Third, in constructing the adjusted model, the DAG was applied to explore the potential confounders and mediating variables between AKI and death, and we carefully accounted for every possible confounder. Finally, the inclusion criteria for patients strictly followed the latest definitions of sepsis and AKI.

Despite these strengths, this study had some drawbacks. First, being a retrospective cohort study, the residual confounders may remain despite having adjusted for many potential confounders. We hypothesize the acute organ failures were mediators between AKI and death as depicted in the DAG, and not included in the models. However, if the failure of organs such as lung, hepatic, or heart play a predominant role in the association between AKI and mortality, or the organ failures were confounders, our results may not evaluate the precise correlation of AKI with mortality. Second, the definition of AKI and baseline creatinine are not possible to be address for the intrinsic nature of this retrospective analysis. Third, we enrolled critically ill patients from ICUs, as such, our findings may not apply to the general patients. Finally, because we focused on sepsis, a common cause of AKI, our results may not be generalizable to patients with AKI attributable to other causes.

Our findings would guide the evaluation of the plausible effect size for future clinical trials regarding the prevention or treatment of AKI.

Conclusions

In two retrospective cohorts of critically ill patients with sepsis, all stage AKI conferred increased risk for hospital mortality, independent of overall severity of illness. Development of AKI was also associated with ICU mortality, hospital and ICU LOS.

Supplemental Information

Supplemental Information 1 Supplemental method and tables

Click here for additional data file.

Supplemental Information 2 Raw data for the Wenzhou cohort

Click here for additional data file.

Supplemental Information 3 Raw data for the Mimic iv cohort

Click here for additional data file.

We wish to thank the intensivists, data managers, and other staff in the participating MIMIC IV Database.

Additional Information and Declarations

Competing Interests

Author Contributions

Human Ethics

Data Availability

The authors declare there are no competing interests.

Zhiyi Wang, Jie Weng, Chan Chen and Shengwei Jin conceived and designed the experiments, analyzed the data, authored or reviewed drafts of the paper, and approved the final draft.

Jinwen Yang, Xiaoming Zhou, Zhe Xu, Ruonan Hou, Zhiliang Zhou and Liang Wang performed the experiments, prepared figures and/or tables, and approved the final draft.

The following information was supplied relating to ethical approvals (i.e., approving body and any reference numbers):

The study was based on existing dataset and was approved by the Ethics Committee of the Second Affiliated Hospital and Yuying Children’s Hospital of Wenzhou Medical University. Informed consent was waived due to retrospective nature of the study. The study was conducted in accordance of the Helsinki Declaration. Committee of the Second Affiliated Hospital and Yuying Children’s Hospital of Wenzhou Medical University.

The following information was supplied regarding data availability:

The raw measurements are available in the Supplemental Files.

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
