# Peer review of "Acute kidney injury-attributable mortality in critically ill patients with sepsis"

_PeerJ, doi:10.7717/peerj.13184_

## Round 0.1 · original submission · Major Revisions

Please pay special attention to clarifying the methodological issues raised by the reviewer's and correct the terminology (use only acute kidney injury, but never acute renal failure - ARF).

Please note that the reviewer's comments also include an annotated pdf file of the manuscript.

·

Basic reporting

The main topic and aim of the authors are clearly explained. The discussion highlights the point of interest of the authors and made meaningful comparisons with current literature.
No major concerns regarding English used, some minor mistakes that can be easily corrected.
Most of the references are meaningful and used properly in the different parts of the manuscript. A couple of reference need to be updated/corrected (line 72 and lines 77-78).
No major concerns regarding the structure of the article and the results shown are relevant to the hypotheses.
Figures and tables are good and clearly highlights the results of the authors. Particularly good is figure 2 to summarize the results.
Raw data regarding the Wenzhou cohort are shared correctly with an excel file available for analysis. Not data regarding. No data regarding the MIMIC IV are shared, to be clarified if it is because too large to be shared with a sample datasheet or for other restrictions.

Experimental design

Research question is well defined, relevant and meaningful. Probably a little expansion of current/previous evidence regarding the association between AKI and mortality should be mentioned already in the introduction. What is actually new in this paper is not the discovery of how AKI increased mortality in critically ill patients with sepsis, but to have clearly measured this association with excess of attributable mortality (and the authors clearly highlight this in lines 214-215 and 235-237 at the end of the discussion).
The methods section and supplemental material explain most of the methodology and statistical analysis. However, some aspect needs to be clarified:
• KDIGO criteria using serum creatine have two time-windows for assessment of AKI: the 48-hour window (increase of serum creatinine>0.3 mg/dL) and the 7-day window (increase of serum creatinine >1.5 time baseline). In the Supplemental Material you mentioned only the 7-day time frame but later in page 3 of the Supplement you showed all the KDIGO criteria, as well the 0.3 mg/dl increase criteria (together with the 1.5-1.9 times baseline serum creatinine). Please clarify if you used also the 0.3 mg/dl increase and in which time window.
• When assessing AKI it is always crucial the definition of "baseline creatinine" since it is what is used to define at the end the presence or not of AKI. This might be tricky sometimes particularly in retrospective studies without predetermined protocolization of serum creatinine/AKI assessment. In the literature different ways are proposed: using the last previous known values of serum creatinine before event index, the mean/median of serum creatinine values in the previous year, admission creatinine at enrollment, back-calculation using MDRD formula from estimated eGFR of 60 ml/min, etc. This should be specified, if there is not space in the methods, in the supplement.
• I will suggest reviewing the paragraph “Primary outcome and additional variables” in the methods section to improve clarity and possibly to divide it in two sections (e.g.: “patients” and “outcomes”).

Validity of the findings

The authors provided important clinical data and they correctly highlighted one of the most important implication of their findings at the end of discussion: “Our findings would guide the evaluation of the plausible effect size for future clinical trials regarding the prevention or treatment of AKI”. I agree with this statement, and I found particularly important the comparison with previous evidence regarding other ICU disease.
The statistical approach used to try to adjust for possible confounders is worth of merit and robust.
The results and the discussion may need just a few minor edits to avoid useless repetition that could allow the authors to have more space to correctly highlight the implications of their findings. I suggest avoiding as much as possible repetition of results clearly and already shown in the tables (e.g.: lines 129-132, 156-159, 204-207, 233-235).
Limitations are clearly stated. If some of the issues I have raised regarding the definition of AKI and baseline creatinine are not possible to be address for the intrinsic nature of this retrospective analysis, they should be stated here.
Conclusions are well stated and limited to supporting results.

Additional comments

I have highlighted in the pdf file of the manuscript some of the issues reported above together with a comments. Please check them.
Regarding the supplement online material:
-second footnote Table S2 and S3: Why not gender? It is not gender is not specified in Table S1 as not considered for Wenzhoud cohort. Or is it just missing in the footnote?
-description of table s4: why EARLY and VALID cohort are mentioned here? Is this a misplacement?
-check references 2, 3 and 4 according to what will be changed/modified in the main text.

Reviewer 2 ·

Basic reporting

Well done

Experimental design

Well done

Validity of the findings

Well done

Additional comments

Well done

Reviewer 3 ·

Basic reporting

Generally well written in professional English with few concerns herein.

1) Note AKI is spelt KAI in line 122.

2) line 283- reference states "and et al" at the end of the author list, whereas all the other reference list all authors.

3) The sentence in line 46 "The excess mortality among patients with AKI, that is, the AKI-related deaths could be avoided without the development of AKI" appears either to be a sentence fragment or a tautology. It is best removed from the manuscript.

4) line 70 "Inclusion criteria for adult patients followed the definition of sepsis-3" is unclear: it appears like a typing error, until I realised it should be "Inclusion criteria for adult patients followed the Third International Consensus Definitions for Sepsis and Septic Shock (Sepsis-3)."

5) In line 212, "The AF of mortality from sepsis was 15% compared to ICU-non-sepsis (Shankar-Hari et al. 2018)" doesn't make sense. It should simply state " "In ICU patients, the AF of mortality from sepsis was 15% (Shankar-Hari et al. 2018)".

6) Several times in the manuscript, the abbreviation (ARF) is used instead of AKI, and ARF is never defined. ARF should be deleted.

Experimental design

There are significant methodological issues:

1) It is unclear why the authors chose to present the same analysis performed on two different cohorts (a larger MIMIC-IV cohort, and a much smaller Wenzhou cohort). Why was this done, instead of combining the Wenzhou patients together with the MIMIC-IV patients to make a single cohort (with analyses simply adjusting for country of location?) This would have cut down the extraneous length of the article significantly. Do the authors have a particular hypothesis about the AKI/sepsis relationship in different countries that they are trying to answer?


2) There is too much relegation of important methodological data to the supplementary material. The directed acyclic graph is unnecessary, and its explaining paragraph in the supplementary material is unclear as the graph does not explicitly label the "indirect pathway". It is important for readers of the primary manuscript to understand the variables taken into account in multivariate models without having to be directed to supplementary material. I suggest that the directed acyclic graph and table S1 be removed, all their relevant data be summarised in text and added to the main manuscript, and a summarised version of the "statistical methods" section in the supplementary material added instead to the main text.

Therefore, the phrase "(variables inclusion criteria are described in Supplementary materials)" in line 137 can also be removed.


3) Line 204 states "The AFAKI is the proportion of deaths attributable to AKI in patients with AKI, whereas the population AFAKI is the proportion of all deaths in the sepsis population attributable to AKI." This is slightly different to the previous definitions given in line 104, and therefore line 204 is both confusing and unnecessary.

Validity of the findings

1) The lines 114 to 121, describing the differences in baseline characteristics between the MIMIC cohort and the Wenzhou cohort, are not backed up by any statistical evidence (e.g. p values) in Table 1. Table 1 only offers p values comparing AKI/non-AKI subgroups within each country's cohort.

2) Line 214 states "Notably, we found that the AF of AKI in patients with sepsis was higher than in other ICU disease states". This is misleading, as it suggests that the current study has included comparator groups of patients that did not have sepsis, which it did not.

3) Similarly lines 208- 211 are written in a confusing manner, with incongruent use of abbreviations. They should state "In one study of critically ill patients with liver cirrhosis, the AFAKI from mild AKI was 25% and the AFAKI from severe AKI was 51% (du Cheyron et al. 2005), whereas the 90-day AFAKI in ICU patients was 8.6%, and population AFAKI was nearly 20% (Vaara et al. 2014) in another study."

---

## Round 0.2 · accepted · Accept

The authors answered all the reviewers' comments and improved their manuscript.

·

Basic reporting

Most of the points presented in my revision have been addressed and clarified.

Experimental design

no comment

Validity of the findings

no comment

Additional comments

no comment

Reviewer 3 ·

Basic reporting

acceptable

Experimental design

My previous issues have been answered.

Validity of the findings

My previous issues have been answered.